# Controllable Preparation of SERS-Active Ag-FeS Substrates by a Cosputtering Technique

**DOI:** 10.3390/molecules24030551

**Published:** 2019-02-02

**Authors:** Ning Ma, Xin-Yuan Zhang, Wenyue Fan, Bingbing Han, Sila Jin, Yeonju Park, Lei Chen, Yongjun Zhang, Yang Liu, Jinghai Yang, Young Mee Jung

**Affiliations:** 1Key Laboratory of Preparation and Applications of Environmental Friendly Materials, Ministry of Education, College of Chemistry, Jilin Normal University, Changchun 130103, China; elva_maning@163.com (N.M.); 15504341371@163.com (W.F.); bingbinghan0106@163.com (B.H.); 2Key Laboratory of Functional Materials Physics and Chemistry, Ministry of Education, College of Physics, Jilin Normal University, Changchun 130103, China; zxyciomp@126.com (X.-Y.Z.); liuyang@jlnu.edu.cn (Y.L.); jhyang1@jlnu.edu.cn (J.Y.); 3Department of Chemistry, Institute for Molecular Science and Fusion Technology, Kangwon National University, Chunchon 24341, Korea; jsira@kangwon.ac.kr (S.J.); yeonju4453@kangwon.ac.kr (Y.P.)

**Keywords:** PSCP templates, cosputtering, SERS, SPR

## Abstract

In this work, we introduced an ordered metal-semiconductor molecular system and studied the resulting surface-enhanced Raman scattering (SERS) effect. Ag-FeS nanocaps with sputtered films of different thicknesses were obtained by changing the sputtering power of FeS while the sputtering power of Ag and the deposition time remained constant. When metallic Ag and the semiconductor FeS are cosputtered, the Ag film separates into Ag islands partially covered by FeS and strong coupling occurs among the Ag islands isolated by FeS, which contributes to the SERS phenomenon. We also investigated the SERS enhancement mechanism by decorating the nanocap arrays produced with different FeS sputtering powers with methylene blue (MB) probe molecules. As the FeS sputtering power increased, the SERS signal first increased and then decreased. The experimental results show that the SERS enhancement can mainly be attributed to the surface plasmon resonance (SPR) of the Ag nanoparticles. The coupling between FeS and Ag and the SPR displacement of Ag vary with different sputtering powers, resulting in changes in the intensity of the SERS spectra. These results demonstrate the high sensitivity of SERS substrates consisting of Ag-FeS nanocap arrays.

## 1. Introduction

In 1974, Fleischmann and his collaborators found for the first time that pyridine on the surface of a rough silver electrode generates a strong Raman signal [1]. The unique advantages of surface-enhanced Raman scattering (SERS) perfectly solve the shortcomings of weak Raman signal strengths and this technique is thus widely used in surface and interface analysis, chemistry, catalysis, biosensing, medical detection and trace analysis [2,3,4]. At present, studies of the nature of the SERS enhancement mechanism are based on the enhancement of the incident electric field, E and the change in the molecular polarizability, α, which correspond to an electromagnetic field enhancement mechanism (EM) [5,6] and a chemical enhancement mechanism (CM) [7,8]. According to classical electromagnetic theory, the Raman scattering intensity is proportional to the square of the molecular dipole moment (*p = αR*). When the frequency of the incident light matches the natural frequency of the electronic oscillation of the surface of the metal nanoparticles (NPs), the amplitude of the oscillation is maximized. This phenomenon is called localized surface plasmon resonance (LSPR) [9]. When an applied electromagnetic field is coupled with surface charge density oscillations, LSPR is excited at the interface between the metal nanofilm and the medium. The position of the LSPR peak is affected by the real part (*ε_1_*) of the dielectric constant of the metal NP and the dielectric constant (*ε_m_*) [10] of the surrounding medium, as well as the size and shape of the particle [11,12]. When a metal is combined with a semiconductor, a new LSPR mode is formed. Resonant coupling between metal NPs and between metal NPs and the semiconductor changes the optical properties. SERS is one of the most important applications of the LSPR effect. EM is an inherent property of metal NPs and is independent of the adsorbed molecules; CM is more complicated than EM. CM occurs via short-range chemical adsorption and is derived from the interactions between the metal substrate and the adsorbed molecules, which change the molecular polarizability [13,14,15,16,17,18,19,20,21,22,23,24,25,26,27]. The SERS effect is achieved by changing the polarizability of the new CT excited state formed by the interactions between the molecule and the substrate [19,28]. EM is generally considered to be much larger than CM but it is difficult to distinguish the two effects strictly by experimental means and the two mechanisms coexist in many systems.

Ag NPs are one of the most common metal materials used for Raman signal enhancement and they have been extensively studied in nanotechnology due to their unique optical properties and inherent SPR properties. However, since Ag is easily oxidized and biologically toxic, the use of this metal is limited [29]. The semiconductor FeS has specific electron-transfer capabilities and good adsorption characteristics as well as a narrow band gap. Optimized FeS shell-encapsulated NPs can be used as a stable SERS-active substrate to avoid the generation of background fluorescence and improve the SERS quality. A FeS layer also reduces the rate of Ag oxidation and increases the stability of the Ag NPs [30]. To date, the SERS mechanism for the interaction between ultranarrow-bandgap semiconductors and metals is still not fully understood. Therefore, designing and fabricating a Ag-FeS composite as a simple and controllable SERS enhancement substrate with semiconductor films of different thickness to regulate the metal SPR is of great importance. The microstructure and properties of nanocomposite film materials are closely related to the preparation process. The properties of the prepared films will be different by changing the substrate bias [31], deposition temperature [32], power or time [31], target composition [33] and N_2_ flow rate [34]. The microstructure and properties can be flexibly controlled by changing the parameters of cosputtering method. It is convenient to study the effects of different power and different sputtering time on the morphology and structure of the film and the effect of SPR on SERS performance.

In our work, a Ag-FeS nanocap array was fabricated by dual target cosputtering; the FeS sputtering power was changed while maintaining a constant deposition time and a constant sputtering power were used for Ag. Thus, Ag-FeS nanocaps with sputtered films of different thicknesses were obtained. The film deposition rate increased as the sputtering power of the FeS target increased. During sputter coating, the deposition rate is proportional to the rate at which atoms escape from the cathode target. In addition, the radio frequency (RF) sputtering power was increased, causing the majority of the sputtered particles to have high energy and these particles will have defects at the impact points. Since the combined energy of these defective regions is higher than that of the adjacent regions, defects are the preferred nucleation sites. Thus, increased sputtering power accelerates film growth and enhances the crystallinity of the resulting film. As the semiconductor FeS is introduced in a large amount in the system, the resulting layer will cover the surface of the Ag particle. At different sputtering powers, the coupling between FeS and Ag and the SPR displacement of Ag change, changing the intensity of the SERS spectra.

## 2. Results and Discussion

The Ag-FeS nanocap arrays were prepared by double-target cosputtering using colloidal sphere arrays of 200-nm polystyrene colloidal particles (PSCPs) as the substrate. The PSCP templates were assembled on silicon wafers by a self-assembly method. Figure 1 and Appendix A show the high and low magnification SEM images of Ag-FeS nanocaps deposited on the PSCP template. These ordered colloidal sphere arrays have a tight, hexagonal symmetrical structure when Ag and FeS are cosputtered on a densely packed PSCP template, forming a rough Ag-FeS nanocap structure. Ag-FeS nanocaps with sputtered films of different thicknesses were obtained by changing the sputtering power of FeS while maintaining a constant deposition time and Ag sputtering power. When the FeS sputtering power was 50 W (Figure 1a), the surfaces of the Ag-FeS nanocaps are still smooth. As the sputtering power increases, small rough NPs appear in individual regions of the PSCP template. When the sputtering power is 80 W (Figure 1d), there are some higher cluster formations on the surfaces of the colloidal spheres; this morphology is due to the increase in FeS sputtering power, which ionizes more Ar into Ar^+^. The number of Fe NPs deposited on the substrate increases and the roughness is maximized. However, when the sputtering power is increased to 90 W, the surface of the sample is smoother due to the sputtered FeS particles and the cumulative agglomeration phenomenon is weakened. This morphological change is mainly due to nucleation and growth processes, as described by film formation and growth theory. The film deposition rate increases as the sputtering power of the FeS target increases. During sputter coating, the deposition rate is proportional to the rate at which atoms escape from the cathode target. In addition, the RF sputtering power is increased, so most of the sputtered particles have high energy and these particles will cause defects at the impact points. Since the combined energy of these defective regions is higher than that of the adjacent regions, the defects become the preferred nucleation sites and accelerated film growth and increase the crystallinity of the resulting film.

The roughness of the base material influences the SPR. In the UV-Vis absorption spectra shown in Figure 2, the absorption peak at approximately 294 nm is attributed to the interband transition of silver [35]. The absorption peak located near 509 nm is attributed to the LSPR between the dipoles derived from the Ag/FeS array [36]. The EM interactions between the dipole and the particle are different, resulting in a change in the peak intensity. The Ag/FeS peaks shift because when the Ag and FeS are deposited, the electronic environment around the array was changed based on the interactions between Ag and FeS. The change in the SPR characteristics is caused by electron transfer [37]. The interparticle coupling is dominant and leads to a shift in the LSPR. As the content of Ag NPs increases, they accumulate on the surface of the base material together with FeS and the in-plane coupling increases, redshifting the LSPR. As the FeS content on the semiconductor increases, outer plane coupling occurs and the in-plane coupling decreases, blueshifting LSPR [38].

The XPS spectra used to determine the phase of the elements and the state of the chemical bonds are given in Figure 3. The peak position at which an offset occurred due to surface static was referenced to contaminant carbon (C 1s = 284.6 eV). The peaks at 711.0 and 724.4 eV in the Fe 2p spectrum shown in Figure 3a are attributed to the presence of Fe-S bonds [39]; the peak at 161.2 eV in the S 2p spectrum shown in Figure 3b is attributed to the FeS characteristic peak of S^2−^, the peak at 162.3 eV is attributed to FeS_2_/FeS [40] and the peaks at 162.9, 163.3 and 163.4 eV correspond to polysulfide. The peaks at 164.5, 168.2 and 168.3 eV indicate the S 2p_3/2_ and S 2p_1/2_ orbits, respectively [28]. The peaks of Ag 3d_5/2_ and 3d_3/2_ appear at 367.9 and 373.9 eV, respectively, in Figure 3c. The presence of C, S, Fe and Ag was confirmed by the XPS full-scan spectrum shown in Figure 3d, indicating the successful preparation of the cosputtered substrate.

To study the SERS activity of the Ag-FeS nanocap film, we chose MB as the probe molecule. Figure 4 shows the Raman spectrum of MB in ethanol; the characteristic Raman bands of MB at 445 and 1618 cm^−1^ attributed to C–C stretching and C–N–C skeleton bending, respectively, can be observed [41] and characteristic bands of ethanol can be observed at 883 and 1067 cm^−1^. However, due to the low concentration, the characteristic MB bands are masked by the Raman scattering generated by the solvent and cannot be clearly observed.

Figure 5 shows the SERS spectra of MB adsorbed on the Ag-FeS module of the system at an excitation wavelength of 514 nm. The distribution of the SERS bands of MB is listed in Table 1 [38]. Compared with the Raman spectrum of MB in ethanol, some bands are better resolved in the SERS spectrum of the Ag-FeS-MB system but the characteristic bands of MB are clear. This distribution indicates that the molecules were well adsorbed onto the substrate material [41,42,43]. Furthermore, at an excitation wavelength of 514 nm, the band at 1618 cm^−1^ splits into two bands at 1594 and 1627 cm^−1^. The displacement and splitting of some of the Raman bands indicates that the MB may be chemisorbed on the surface of the Ag-FeS nanocap film [44]. The peaks marked with triangles in the Figure 5a at 520 and 920–1000 cm^−1^ are assigned to the Si peaks [45]. Figure 2 shows that the SPR absorption peak is closest to 514 nm at a FeS sputtering power of 80 W. Due to the resonance effect between the substrate and the laser at the excitation wavelength of 514 nm, the SERS spectrum of the substrate obtained with a FeS sputtering power of 80 W is the strongest, which consistent with the results of the UV-Vis spectra. We selected the characteristic MB band at 1627 cm^−1^ and calculated the ratios of the Raman intensity as the sputtering power increased. When the FeS sputtering power is 80 W, the intensity of the MB signals reached their maximum. Based on the trend in the intensity of the band at 1627 cm^−1^, the maximum signal at 80 W is more intense that the corresponding signals obtained with other powers. An increase in power decreases the intensity of the signal due to the increased content of the semiconductor, which affects the electromagnetic field enhancement, so the SERS intensity gradually decreases after reaching the maximum value.

The intensities of the SERS signals are closely related to the size of the nanostructures, the surface roughness, the surrounding electrolyte environment and ‘hot spots.’ When Ag and FeS are cosputtered, the Ag film is separated into Ag islands partially covered by FeS and strong coupling occurs between the Ag islands isolated by FeS, which contributes to the SERS phenomenon. NPs optimized with semiconductor FeS shells can be used as a stable SERS-active substrate to avoid background fluorescence [46], which improves the quality of the SERS spectra. The deposited Ag film will gradually degrade and a FeS shell of the appropriate thickness as a metal deposition mask also reduces the rate of Ag oxidation, increases the stability of Ag and produces a greater SERS effect. However, as the distance from the surface of the metal to the outer surface of the FeS shell increases, the intensity of the electromagnetic field exponentially decrease and an excessively thick FeS shell will mask the surface plasmon band, which hinders the generation of SERS. The change in the SERS intensity of the Ag-FeS nanocap array is mainly due to the matching of the plasmon resonance peak with the excitation wavelength and the difference in the structural morphology. Therefore, sputtering FeS at different powers changes the surface roughness, making the plasma oscillation adjustable within a certain range and changing the SERS intensity. At an excitation wavelength of 514 nm, when Ag-FeS nanocap arrays are deposited on a 200-nm PSCP template, the sample prepared at a sputtering power of 80 W has the largest surface roughness, which greatly increases the number of ‘hot spot’ regions, promoting the localization of electrons and significantly enhancing the SERS signal.

After obtaining the enhanced Raman signals of MB on the Ag/FeS array, we tried to estimate the surface enhancement factor on the array. The SERS enhancement factor (EF) is an important physical quantity for the quantitative comparison of substrate SERS activity, and it is the most important parameter in studies on SERS mechanisms. The EF was calculated by randomly selecting 8 points from an Ag/FeS array of 80 W sputtering power using an equation [47]:(1)$$EF=\frac{I_{SERS}}{I_{bulk}}\times \frac{N_{bulk}}{N_{SERS}}$$
where *I_SERS_* and *I_bulk_* are the SERS intensity of the band at 1627 cm^−1^ (assigned to MB absorbed on the Ag/FeS arrays) and the Raman intensity of the band at 1618 cm^−1^ (assigned to solid MB), respectively. *N_bulk_ = A_laser_hcN_A_*, where *N_bulk_* is the number of molecules in the solid material under illumination, *A_laser_* is the area of the focused laser spot (1 µm in diameter) for Raman scanning, the confocal depth (h) of the Renishaw Micro-Raman spectrometer with 514 nm laser excitation is 19 µm, *C* is the density of the MB molecules, *N_A_* is Avogadro’s constant, and *N_SERS_* is the average number of adsorbed molecules in the scattering volume for the SERS experiments. *N_SERS_ = A_laser_N_d_A_N_/**δ*, where *N_d_* is the number density of 200-nm PS, *A_N_* is the half surface area of one 200-nm PS, and *δ* is the surface area occupied by a single molecule of MB adsorbed on the substrate, which is estimated to be 0.75 nm^2^. *N_bulk_*/*N_SERS_* can be calculated to be 6.9 × 10^5^. Therefore, EF of the cosputtered Ag/FeS arrays can be calculated to be 1.8 × 10^6^. Thus, the SERS effect of the prepared Ag/FeS array substrate shows that the substrate has a good Raman enhancement effect, and the enhancement factor is as high as 1.8 × 10^6^. The substrate prepared by this method has potential value in studies on SERS.

## 3. Materials and Methods

### 3.1. Chemicals and Materials

P-type Si (100) wafers with a size of 200 µm were obtained from Hefei Kejing Materials Technology Co., Ltd. (Hefei, China) Monodisperse polystyrene colloidal particles (PSCPs) (200 nm, 10 wt% aqueous solution) were purchased from Bangs Laboratories Inc. (Shenzhen, China). Sodium dodecyl sulphate (SDS) and methylene blue (MB) were purchased from Sigma-Aldrich Co., Ltd. (Shanghai, China). NH_4_OH (25%) and H_2_O_2_ (30%) were purchased from Sinopharm Chemical Reagent Co., Ltd. (Shanghai, China). Ag and FeS targets with purities of 99.99% were obtained from Beijing Tianrui Technology Co., Ltd. (Beijing, China). All chemicals were used without further purification. Ethanol and deionized ultrapure water (18.25 MΩ cm^−1^) were used throughout the study.

### 3.2. Preparation of Two-Dimensional (2D) Ordered PSCP Templates

We prepared two-dimensional (2D) ordered 200-nm PSCP templates by a self-assembly technique. Si wafers (with dimensions of 2.0 × 2.0 cm) were immersed in a mixture of NH_4_OH, H_2_O_2_ and H_2_O in a volume ratio of 1:2:6 and heated at 300 °C for 5 min. The wafers were then ultrasonically washed three times with deionized water and ethanol. The treated Si wafers were immersed in sodium lauryl sulphate for 24 h to make the surfaces hydrophilic. Subsequently, ethanol and 200-nm PSCPs were uniformly mixed at a 1:1 volume ratio and dropped on a large piece of Si wafer. The Si wafer was immersed in water at a tilt angle of 45° and the arrays on the Si wafer were transferred onto the surface of the water. A drop of sodium lauryl sulphate solution was then added to the surface and the PSCPs formed ordered monolayer films on the surface of the water. Finally, the arrays were removed by washing the Si wafer and completely dried in air by static natural evaporation. Tightly packed monolayer ordered arrays were formed on the surface of the Si substrate and stored at room temperature for further use.

### 3.3. Fabrication of Ag/FeS Arrays on PSCP Templates

The FeS and Ag targets were simultaneously deposited onto the 200-nm PSCP templates by a magnetron sputtering system (ATC 1800-F, AJA, North Scituate, MA, USA). The processed Si wafers with the PSCP templates were loaded into the cavity of the magnetron sputtering system; the Ag target and the FeS target were loaded into the magnetic target and the nonmagnetic target, respectively. The target angle was 10° and the distance between the substrate and the target was 20 cm. Then, the main magnetron sputtering chamber was evacuated and the background pressure was lower than 1.0 × 10^−6^ mTorr when starting the experiment. The Ar gas flow rate was 8.7 sccm (standard cubic centimetres per minute) and the working pressure was on the order of 5.6 × 10^−3^ mTorr. The sputtering power of Ag was fixed at 5 W during the cosputtering process, the FeS particle size was adjusted by changing the sputtering power of FeS (50, 60, 70, 80 and 90 W) and the sputtering time was 300 s. A schematic of the preparation process of PSCP@Ag-FeS is shown in Scheme 1.

### 3.4. Characterization and SERS Measurements

To study the microstructure and morphology of the PSCP@Ag-FeS arrays, we obtained images using scanning electron microscopy (SEM, JEOL 6500F) (JEOL, Tokyo, Japan) at an accelerating voltage of 200 kV. UV-Vis absorption spectra of the PSCP@Ag-FeS arrays were recorded on a Shimadzu UV-3600 UV-Vis spectrometer (Shimadzu, Kyoto, Japan). To measure the elemental composition and chemical state of PSCP@Ag-FeS, X-ray photoelectron spectroscopy (XPS) measurements were performed using a Thermo Scientific ESCALAB 250Xi A1440 system (Thermo Fisher Scientific, Waltham, MA, USA) with Al Kα as the X-ray source and the XPS spectra were referenced to carbon (C 1s = 284.6 eV). The Raman spectra were measured utilizing Ar^+^ ion laser excitation in a Renishaw Raman System 2000 spectrometer (Renishaw, London, UK) with a confocal microscope. The PSCP@Ag-FeS ordered arrays prepared by the above cosputtering method were immersed in MB solution at a concentration of 10^−3^ mol/L for 3 h and the Raman spectra of the PSCP@Ag-FeS-MB assemblies were acquired at an excitation wavelength of 514 nm. Throughout the experiment, all chemicals were used as received without further purification.

## 4. Conclusions

In summary, we prepared a SERS-active substrate by combining self-assembly techniques and magnetron cosputtering of Ag-FeS metal-semiconductor composites on a 200-nm PSCP template. Ag-FeS nanocaps with sputtered films of different thicknesses were obtained by changing the sputtering power of FeS while maintaining the same deposition time and Ag sputtering power. The deposition rate of the film increased as the sputtering power of the FeS target increased. During sputter coating, the deposition rate is proportional to the rate at which atoms escape from the cathode target. In addition, the RF sputtering power is increased, resulting in most of the sputtered particles have high energy and these particles will cause defects at impact points. Since the combined energy of these defective regions is higher than that of the adjacent regions, the defects become the preferred nucleation sites and accelerate film growth and increase the crystallinity of the resulting films. When a large amount of FeS is introduced into the system, the layer will cover the surface of the Ag metal. The coupling between FeS and Ag and the SPR displacement of Ag changes at different sputtering powers, changing the intensity of the SERS spectra, and the enhancement factor is as high as 1.8 × 10^6^. These results clearly demonstrate the high sensitivity of the Ag-FeS nanocap array as a SERS substrate.

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
