# Peer review of "Controllable Preparation of SERS-Active Ag-FeS Substrates by a Cosputtering Technique"

_molecules, 2019, doi:10.3390/molecules24030551_

Round 1

Reviewer 1 Report

This manuscript is about the preparation of SERS active Ag-FeS arrays by cosputtering. The film thickness as controlled by by changing the sputtering power of FeS. The SERS properties of prepared arrays were tested using standard dye Methylene blue. It is claimed that "The coupling between FeS and Ag and the SPRdisplacement of Ag change at different sputtering powers, resulting in changes in the SERS spectral intensity without any proper evidence. Overall, this is another routine addition to large number of SERS substrates known in literature. It is technically fine. However, I think the presentation can be improved.

Comments:

The scheme for the fabrication procedure should be as Fig.1. Simple "Scheme 1. The fabrication process of the ordered Ag/FeS arrays" is written, it has to be elaborated. Readers should be able to understand from the caption.

2. Fig.1: Simply SEM images are shown, but it is not clear what is what. Where are Ag and FeS, how they can be identified? Any EDX mapping?

3. The origin of UV vis absorption spectra is unclear. What are the peak correspond to FeS and Ag? Explain the peaks shown in the spectra. Ag NPs exhibit extinction at 400 nm Nanoscale 5 (10), 4355-4361), which is not present here. The peaks ~ 500 nm could be due to SPR coupling of Ag NPs or due to anisotropic morphology (JACS 138 (36), 11453-11456). Here none of these are clear. What is the size of Ag NPs? distance between them is important for SPR coupling? even in the case of strong coupling, they should still exhibit peak at 400 nm. I think authors should discuss all these with proper literature servey.

4. I don't understand the purpose of SERS study presented here. What is the enhancement factor? detection sensitivity? better than reported? what are the advantages of these substrates over many reported substrates. Discuss the clear motivation in the introduction and see how far the purpose achieved from these results. To be honest, at present this study is too routine and outdated.

Author Response

All the replied are written in boldface.

We appreciate reviewer’s comments on our manuscript which have made us to revise the manuscript significantly.

Response on comments of Reviewer 1

Comments:

This manuscript is about the preparation of SERS active Ag-FeS arrays by cosputtering. The film thickness as controlled by by changing the sputtering power of FeS. The SERS properties of prepared arrays were tested using standard dye Methylene blue. It is claimed that "The coupling between FeS and Ag and the SPR displacement of Ag change at different sputtering powers, resulting in changes in the SERS spectral intensity without any proper evidence. Overall, this is another routine addition to large number of SERS substrates known in literature. It is technically fine. However, I think the presentation can be improved.

The scheme for the fabrication procedure should be as Fig.1. Simple "Scheme 1. The fabrication process of the ordered Ag/FeS arrays" is written, it has to be elaborated. Readers should be able to understand from the caption.

Responses: Based on the Reviewer’s comments, we have expanded the caption. To clarify the process and make the figure more intuitive, we have revised Scheme 1.

The following figure was added to the revised manuscript as Scheme 1.

Scheme 1. A large, hydrophilic silicon wafer with the ethanol/PSCP mixture was immersed in water at a tilt angle of 45°, and the arrays on the Si wafer were transferred onto the surface of the water. Then, the arrays were removed by washing the 2.0×2.0 cm Si wafer, and they were completely dried in air by static natural evaporation. The FeS and Ag targets were simultaneously deposited onto the 200-nm PSCP templates by a magnetron sputtering system. The SERS spectrum of the PSCP@Ag-FeS-MB assemblies was collected using an excitation wavelength of 514 nm.

2. Fig.1: Simply SEM images are shown, but it is not clear what is what. Where are Ag and FeS, how they can be identified? Any EDX mapping?

Responses: Based on the Reviewer’s comments, we have revised the SEM images.

The following figure was added to the revised manuscript as Figure 1.

Figure 1. SEM images of the ordered Ag-FeS arrays, which were prepared by cosputtering Ag with a constant sputtering power (5 W) and FeS with varied sputtering powers for 300 s on the PSCP templates. The sputtering powers of FeS were (a) 50 W, (b) 60 W, (c) 70 W, (d) 80 W, and (e) 90 W; sputtered pure FeS (f); and pure Ag (g) for 300 s.

3. The origin of UV vis absorption spectra is unclear. What are the peak correspond to FeS and Ag? Explain the peaks shown in the spectra. Ag NPs exhibit extinction at 400 nm Nanoscale 5 (10), 4355-4361), which is not present here. The peaks ~ 500 nm could be due to SPR coupling of Ag NPs or due to anisotropic morphology (JACS 138 (36), 11453-11456). Here none of these are clear. What is the size of Ag NPs? distance between them is important for SPR coupling? even in the case of strong coupling, they should still exhibit peak at 400 nm. I think authors should discuss all these with proper literature servey.

Responses: Based on the Reviewer’s comments, we have assigned the peaks of the UV-Vis spectrum, discussed the reasons for their occurrence and added the appropriate references for discussion.

The following sentences were added to the revised manuscript.

The roughness of the base material influences the SPR. In the UV-Vis absorption spectra shown in Figure 2, the absorption peak at approximately 294 nm is attributed to the interband transition of silver [31]. The absorption peak located near 509 nm is attributed to the LSPR between the dipoles derived from the Ag/FeS array [32]. The EM interactions between the dipole and the particle are different, resulting in a change in the peak intensity. The Ag/FeS peaks shift because when the Ag and FeS are deposited, the electronic environment around the array was changed based on the interactions between Ag and FeS. The change in the SPR characteristics is caused by electron transfer [33]. The interparticle coupling is dominant and leads to a shift in the LSPR. As the content of Ag NPs increases, they accumulate on the surface of the base material together with FeS, and the in-plane coupling increases, redshifting the LSPR. As the FeS content on the semiconductor increases, outer plane coupling occurs and the in-plane coupling decreases, blueshifting LSPR [34].”

We have added the following new references in the revised manuscript.

References

31.        Ganeev, R.; Ryasnyanskiy, A.I.; Stepanov, A.L.; Usmanov, T. Saturated absorption and nonlinear refraction of silicate glasses doped with silver nanoparticles at 532 nm. Opt. Quant. Electron 2004, 36, 949-960. https://doi.org/10.1007/s11082-004-3392-x.

32.        Zhang, T.; Sun, Y.; Huang, L.; Li, L.H.; Liu, G.; Zhang, X.; Lyu, X.; Cai, W.; Li, Y. Periodic Porous Alloyed Au-Ag Nanosphere Arrays and Their Highly SensitiveSERS Performance with Good Reproducibility and High Density of Hotspots. ACS Appl. Mater. Inter. 2018, 10, 9792-9801. https://doi.org/10.1021/acsami.7b17461.

33.        Singh, A.N.; Devnani, H.; Jha, S.; Ingole, P.P. Fermi Level Equilibration of Ag and Au Plasmonic Metal Nanoparticles supported on Graphene Oxide. Phys. Chem. Chem. Phys. 2018, 20, 26719-26733. https://doi.org/10.1039/c8cp05170d.

34.        Ghodselahi, T.; Neishaboorynejad, T.; Arsalani, S. Fabrication LSPR sensor chip of Ag NPs and their biosensor application based on inter-particle coupling. Appl. Surf. Sci. 2015, 343, 194-201. https://doi.org/10.1016/j.apsusc.2015.01.219.

I don't understand the purpose of SERS study presented here. What is the enhancement factor? detection sensitivity? better than reported? what are the advantages of these substrates over many reported substrates. Discuss the clear motivation in the introduction and see how far the purpose achieved from these results. To be honest, at present this study is too routine and outdated.

Responses: Based on the Reviewer’s comments, we have added a section describing the calculation of the enhancement factor to the “Supporting Information”.

The following sentences were added to the “Supporting Information”.

After obtaining the enhanced Raman signals of MB on the Ag/FeS array, we tried to estimate the surface enhancement factor on the array. The SERS enhancement factor (EF) is an important physical quantity for the quantitative comparison of substrate SERS activity, and it is the most important parameter in studies on SERS mechanisms. The EF was calculated from 8 randomly selected points on the Ag/FeS array using equation [1]:

                        (1)

where ISERS and Ibulk are the SERS intensity of the band at 1627 cm−1 (assigned to MB absorbed on the Ag/FeS arrays) and the Raman intensity of the band at 1618 cm−1 (assigned to solid MB), respectively. Nbulk=AlaserhcNA, where Nbulk is the number of molecules in the solid material under illumination, Alaser is the area of the focused laser spot (1 µm in diameter) for Raman scanning, the confocal depth (h) of the Renishaw Micro-Raman spectrometer with 514-nm laser excitation is 19 µm, C is the density of the MB molecules, NA is Avogadro’s constant, and NSERS is the average number of adsorbed molecules in the scattering volume for the SERS experiments. NSERS=AlaserNdAN, where Nd is the number density of 200-nm PS, AN is the half surface area of one 200-nm PS, and δ is the surface area occupied by a single molecule of MB adsorbed on the substrate, which is estimated to be 0.75 nm2. Nbulk/NSERS can be calculated to be 6.9×105. Therefore, EF of the cosputtered Ag/FeS arrays can be calculated to be 1.8×106. Thus, the SERS effect of the prepared Ag/FeS array substrate shows that the substrate has a good Raman enhancement effect, and the enhancement factor is as high as 1.8×106. The substrate prepared by this method has potential value in studies on SERS.

The following new reference was added in the “Supporting Information”.

Reference

1.          Green, M.; Liu, F.M.   SERS Substrates Fabricated by Island Lithography: The Silver/Pyridine System. J. Phys. Chem. B 2003, 107, 13015-13021. https://doi.org/10.1021/jp030751y.

Reviewer 2 Report

The paper provides high potential and suggests innovative approach for SERS-substrate fabrication from a combination of noble metal-semiconductor nanomaterial.

The results sections includes specific investigations for this type of SERS-substrate and is thoroughly elaborated. However, I consider the tested molecule very popular and omnipresent in SERS research area with far better results. Is the enhancement factor determined and not shown? Please include tests for EF determination. In other words, the work is valuable if comparable with other nanostructures' performance proposed in literature.

The conclusion “These results clearly demonstrate the high sensitivity of the Ag‐FeS nanocap array SERS substrate” is not based on results concerning the enhancement factor. There is potential for SERS applications by using this SERS-active substrate, but not proven.

Author Response

All the replied are written in boldface.

We appreciate reviewer’s comments on our manuscript which have made us to revise the manuscript significantly.

Response on comments of Reviewer 2

Comments:

The paper provides high potential and suggests innovative approach for SERS-substrate fabrication from a combination of noble metal-semiconductor nanomaterial. The results sections includes specific investigations for this type of SERS-substrate and is thoroughly elaborated. However, I consider the tested molecule very popular and omnipresent in SERS research area with far better results. Is the enhancement factor determined and not shown? Please include tests for EF determination. In other words, the work is valuable if comparable with other nanostructures' performance proposed in literature. The conclusion “These results clearly demonstrate the high sensitivity of the AgFeS nanocap array SERS substrate” is not based on results concerning the enhancement factor. There is potential for SERS applications by using this SERS-active substrate, but not proven.

Responses: Based on the Reviewer’s comments, we have added a section describing the calculation of the enhancement factor to the “Supporting Information”.

The following sentences were added to the “Supporting Information”.

“After obtaining the enhanced Raman signals of MB on the Ag/FeS array, we tried to estimate the surface enhancement factor on the array. The SERS enhancement factor (EF) is an important physical quantity for the quantitative comparison of substrate SERS activity, and it is the most important parameter in studies on SERS mechanisms. The EF was calculated from 8 randomly selected points on the Ag/FeS array using equation [1]:

                        (1)

where ISERS and Ibulk are the SERS intensity of the band at 1627 cm−1 (assigned to MB absorbed on the Ag/FeS arrays) and the Raman intensity of the band at 1618 cm−1 (assigned to solid MB), respectively. Nbulk=AlaserhcNA, where Nbulk is the number of molecules in the solid material under illumination, Alaser is the area of the focused laser spot (1 µm in diameter) for Raman scanning, the confocal depth (h) of the Renishaw Micro-Raman spectrometer with 514-nm laser excitation is 19 µm, C is the density of the MB molecules, NA is Avogadro’s constant, and NSERS is the average number of adsorbed molecules in the scattering volume for the SERS experiments. NSERS=AlaserNdAN, where Nd is the number density of 200-nm PS, AN is the half surface area of one 200-nm PS, and δ is the surface area occupied by a single molecule of MB adsorbed on the substrate, which is estimated to be 0.75 nm2. Nbulk/NSERS can be calculated to be 6.9×105. Therefore, EF of the cosputtered Ag/FeS arrays can be calculated to be 1.8×106. Thus, the SERS effect of the prepared Ag/FeS array substrate shows that the substrate has a good Raman enhancement effect, and the enhancement factor is as high as 1.8×106. The substrate prepared by this method has potential value in studies on SERS.”

The following new reference was added in the “Supporting Information”.

Reference

1.          Green, M.; Liu, F.M.   SERS Substrates Fabricated by Island Lithography: The Silver/Pyridine System. J. Phys. Chem. B 2003, 107, 13015-13021. https://doi.org/10.1021/jp030751y.

Reviewer 3 Report

Comments:

In this manuscript, the authors designed Ag-FeS nanocaps with different sputtered film thicknesses were obtained by changing the sputtering power of FeS while the sputtering power of Ag and deposition time remained constant. The proposed substrate exhibited the good SERS enhancement. The authors investigated the SERS enhancement mechanism by decorating methylene blue (MB) probe molecules onto nanocap arrays produced with different FeS sputtering powers. The experimental results show that the SERS enhancement is mainly attributed to the coupling between FeS and Ag and the surface plasmon resonance (SPR) displacement of Ag change at different sputtering powers.

This manuscript is publishable subject to minor revision. Some specific issues should be reconsidered:

1) In the introduction part, the authors make the following statement: “Since the discovery of the Raman scattering effect in 1928, Raman scattering and its applications, especially surface-enhanced Raman scattering (SERS), have developed rapidly.” Please cite related reference.

2) In the introduction part, the authors make the following statement: “The LSPR peak position is affected by the real part ε1 of the dielectric constant of the metal nanoparticle and the dielectric constant εm of the surrounding medium, as well as the size and shape of the particle.” Please check ε1 and εm format and mark them as subscript formats.

3) The authors should explain clearly of the Scheme 1 in the “Results and discussion”.

4) Please increase the resolution of the Scheme 1.

5) …… FeS shell will increase the distance of metal ions to the outer surface, exponentially decrease the intensity of the electromagnetic field…….”, The authors mean that the metal ions transfer to the outer surface? Please check it.

6) Some most recently literature related to this topic should be commented in this paper.

Author Response

All the replied are written in boldface.

We appreciate reviewer’s comments on our manuscript which have made us to revise the manuscript significantly.

Response on comments of Reviewer 3

Comments:

In this manuscript, the authors designed Ag-FeS nanocaps with different sputtered film thicknesses were obtained by changing the sputtering power of FeS while the sputtering power of Ag and deposition time remained constant. The proposed substrate exhibited the good SERS enhancement. The authors investigated the SERS enhancement mechanism by decorating methylene blue (MB) probe molecules onto nanocap arrays produced with different FeS sputtering powers. The experimental results show that the SERS enhancement is mainly attributed to the coupling between FeS and Ag and the surface plasmon resonance (SPR) displacement of Ag change at different sputtering powers.

This manuscript is publishable subject to minor revision. Some specific issues should be reconsidered:

In the introduction part, the authors make the following statement: “Since the discovery of the Raman scattering effect in 1928, Raman scattering and its applications, especially surface-enhanced Raman scattering (SERS), have developed rapidly.” Please cite related reference.

Responses: Based on the Reviewer’s comments, we have appropriately edited the introduction and introduced references in the corresponding locations.

In the introduction part, the authors make the following statement: “The LSPR peak position is affected by the real part ε1 of the dielectric constant of the metal nanoparticle and the dielectric constant εm of the surrounding medium, as well as the size and shape of the particle.” Please check ε1 and εm format and mark them as subscript formats.

Responses: Based on the Reviewer’s comments, we have corrected this sentence as follows.

“The position of the LSPR peak is affected by the real part (ε1) of the dielectric constant of the metal NP and the dielectric constant (εm) [10] of the surrounding medium, as well as the size and shape of the particle [11,12].”

The authors should explain clearly of the Scheme 1 in the “Results and discussion”.

Responses: Based on the Reviewer’s comments, we have expanded the caption. To clarify the process and make the figure more intuitive, we have revised Scheme .

The following figure was added to the revised manuscript as Scheme 1.

Scheme 1. A large, hydrophilic silicon wafer with the ethanol/PSCP mixture was immersed in water at a tilt angle of 45°, and the arrays on the Si wafer were transferred onto the surface of the water. Then, the arrays were removed by washing the 2.0×2.0 cm Si wafer, and they were completely dried in air by static natural evaporation. The FeS and Ag targets were simultaneously deposited onto the 200-nm PSCP templates by a magnetron sputtering system. The SERS spectrum of the PSCP@Ag-FeS-MB assemblies was collected using an excitation wavelength of 514 nm.

Please increase the resolution of the Scheme 1.

Responses: Based on the Reviewer’s comments, we have increased the resolution of Scheme 1.

The following figure was added to the manuscript as Scheme 1.

Scheme 1. A large, hydrophilic silicon wafer with the ethanol/PSCP mixture was immersed in water at a tilt angle of 45°, and the arrays on the Si wafer were transferred onto the surface of the water. Then, the arrays were removed by washing the 2.0×2.0 cm Si wafer, and they were completely dried in air by static natural evaporation. The FeS and Ag targets were simultaneously deposited onto the 200-nm PSCP templates by a magnetron sputtering system. The SERS spectrum of the PSCP@Ag-FeS-MB assemblies was collected using an excitation wavelength of 514 nm.

“……FeS shell will increase the distance of metal ions to the outer surface, exponentially decrease the intensity of the electromagnetic field…….”, The authors mean that the metal ions transfer to the outer surface? Please check it.

Responses: Based on the Reviewer’s comments, we have revised this sentence.

The following sentences were corrected in the revised manuscript.

However, as the distance from the surface of the metal to the outer surface of the FeS shell increases, the intensity of the electromagnetic field exponentially decrease, and an excessively thick FeS shell will mask the surface plasmon band, which hinders the generation of SERS.

Some most recently literature related to this topic should be commented in this paper.

Responses: Based on the Reviewer’s comments, we have cited the appropriate references in the revised manuscript.

The following new references were added in the revised manuscript.

2.          Dai, Z.G.; Xiao, X.H.; Wu, W.; Zhang, Y.P.; Liao, L.; Guo, S.S.; Ying, J.J.; Shan, C.X.; Sun, M.T.; Jiang, C.Z. Plasmon-driven reaction controlled by the number of graphenelayers and localized surface plasmon distribution during optical excitation. Light-SCI Appl. 2015, 4. https://doi.org/10.1038/lsa.2015.115.

3.          Rusnati, M.; Sala, D.; Orro, A.; Bugatti, A.; Trombetti, G.; Cichero, E.; Urbinati, C.; Di Somma, M.; Millo, E.; Galietta, L.J.V.; Milanesi, L.; Fossa, P.; D'Ursi, P. Speeding Up the Identification of Cystic Fibrosis Transmembrane Conductance Regulator-Targeted Drugs: An Approach Based on Bioinformatics Strategies and Surface Plasmon Resonance. Molecules 2018, 23, 120. https://doi.org/10.3390/molecules23010120.

4.          Meuser, M.E.; Murphy, M.B.; Rashad, A.A.; Cocklin, S. Kinetic Characterization of Novel HIV-1 Entry Inhibitors: Discovery of a Relationship between Off-Rate and Potency. Molecules 2018, 23, 1940. https://doi.org/10.3390/molecules23081940.

Reviewer 4 Report

After reading the paper entitled "Controllable preparation of SERS-active Ag-FeS substrate by co-sputtering technique", I think that this latter is not very suitable for this type of journal. Moreover, this paper is badly written and presented. Also it misses some important informations in this paper (see the different points below). You will find below a few revisions which should be addressed.

1-Please can your reduce the introduction because it is too long? For instance, you can suppress the definitions of LSPR and SPR.

2-I think that the outline will be better by putting the part 3 before the part 2 "Results & Discussions". Thus, your paper would be more readable.

3- Please also add a SEM image on a larger zone (Ex: scale bar = 20 microns) because actually these are only zooms with a scale bar of 100 nm in order to see your uniformity of your assembly.

4-In your conclusion, you speak about a high sensitivity obtained with Ag-FeS nanocaps. A high sensitivity in terms of analyte concentration (limit of detection) ?? For instance, can you add the calculation the enhancement factor for this type of SERS substrate? Also, can you explain the SERS enhancement by using the absorbance spectra?

5-The scheme 1 is not very readable. Can you improve it in terms of resolution?

6- In your SERS spectra, other Raman shifts are observed? These are other Raman peaks of MB molecules)? Or the Raman peaks of Si, or others.

7-Please read 2 or 3 times your manuscript, because it remain some spelling errors.

Author Response

All the replied are written in boldface.

We appreciate reviewer’s comments on our manuscript which have made us to revise the manuscript significantly.

Response on comments of Reviewer 4

Comments:

After reading the paper entitled "Controllable preparation of SERS-active Ag-FeS substrate by co-sputtering technique", I think that this latter is not very suitable for this type of journal. Moreover, this paper is badly written and presented. Also it misses some important informations in this paper (see the different points below). You will find below a few revisions which should be addressed.

Please can your reduce the introduction because it is too long? For instance, you can suppress the definitions of LSPR and SPR.

Responses: Based on the Reviewer’s comments, we have appropriately edited the introduction to the manuscript.

I think that the outline will be better by putting the part 3 before the part 2 "Results & Discussions". Thus, your paper would be more readable.

Responses: Based on the Reviewer’s comments, we have put part 3 before part 2 in the revised manuscript.

3. Please also add a SEM image on a larger zone (Ex: scale bar = 20 microns) because actually these are only zooms with a scale bar of 100 nm in order to see your uniformity of your assembly.

Responses: Based on the Reviewer’s comments, we have revised the SEM images and increased the magnification of the SEM images as much as possible.

The following figure was added to the revised manuscript as Figure 1.

Figure 1. SEM images of the ordered Ag-FeS arrays, which were prepared by cosputtering Ag with a constant sputtering power (5 W) and FeS with varied sputtering powers for 300 s on the PSCP templates. The sputtering powers of FeS were (a) 50 W, (b) 60 W, (c) 70 W, (d) 80 W, and (e) 90 W; sputtered pure FeS (f); and pure Ag (g) for 300 s.

4. In your conclusion, you speak about a high sensitivity obtained with Ag-FeS nanocaps. A high sensitivity in terms of analyte concentration (limit of detection) ?? For instance, can you add the calculation the enhancement factor for this type of SERS substrate? Also, can you explain the SERS enhancement by using the absorbance spectra?

Responses: Based on the Reviewer’s comments, we have added a section describing the calculation of the enhancement factor to the “Supporting Information”. Figure 2 shows that the SPR absorption peak is closest to 514 nm at a FeS sputtering power of 80 W. Due to the resonance effect between the substrate and the laser at an excitation wavelength of 514 nm, the SERS spectrum of the substrate obtained with a FeS sputtering power of 80 W is the strongest, which consistent with the results of the UV-Vis spectra.

The following sentences were added to the “Supporting Information”.

“After obtaining the enhanced Raman signals of MB on the Ag/FeS array, we tried to estimate the surface enhancement factor on the array. The SERS enhancement factor (EF) is an important physical quantity for the quantitative comparison of substrate SERS activity, and it is the most important parameter in studies on SERS mechanisms. The EF was calculated from 8 randomly selected points on the Ag/FeS array using equation [1]:

                        (1)

where ISERS and Ibulk are the SERS intensity of the band at 1627 cm−1 (assigned to MB absorbed on the Ag/FeS arrays) and the Raman intensity of the band at 1618 cm−1 (assigned to solid MB), respectively. Nbulk=AlaserhcNA, where Nbulk is the number of molecules in the solid material under illumination, Alaser is the area of the focused laser spot (1 µm in diameter) for Raman scanning, the confocal depth (h) of the Renishaw Micro-Raman spectrometer with 514-nm laser excitation is 19 µm, C is the density of the MB molecules, NA is Avogadro’s constant, and NSERS is the average number of adsorbed molecules in the scattering volume for the SERS experiments. NSERS=AlaserNdAN, where Nd is the number density of 200-nm PS, AN is the half surface area of one 200-nm PS, and δ is the surface area occupied by a single molecule of MB adsorbed on the substrate, which is estimated to be 0.75 nm2. Nbulk/NSERS can be calculated to be 6.9×105. Therefore, EF of the cosputtered Ag/FeS arrays can be calculated to be 1.8×106. Thus, the SERS effect of the prepared Ag/FeS array substrate shows that the substrate has a good Raman enhancement effect, and the enhancement factor is as high as 1.8×106. The substrate prepared by this method has potential value in studies on SERS.”

The following new reference was added in the “Supporting Information”.

Reference

1.          Green, M.; Liu, F.M.   SERS Substrates Fabricated by Island Lithography: The Silver/Pyridine System. J. Phys. Chem. B 2003, 107, 13015-13021. https://doi.org/10.1021/jp030751y.

The scheme 1 is not very readable. Can you improve it in terms of resolution?

Responses: Based on the Reviewer’s comments, we have improved the resolution of this figure and expanded the caption.

The following figure was added to the revised manuscript as Scheme 1.

Scheme 1. A large, hydrophilic silicon wafer with the ethanol/PSCP mixture was immersed in water at a tilt angle of 45°, and the arrays on the Si wafer were transferred onto the surface of the water. Then, the arrays were removed by washing the 2.0×2.0 cm Si wafer, and they were completely dried in air by static natural evaporation. The FeS and Ag targets were simultaneously deposited onto the 200-nm PSCP templates by a magnetron sputtering system. The SERS spectrum of the PSCP@Ag-FeS-MB assemblies was collected using an excitation wavelength of 514 nm.

In your SERS spectra, other Raman shifts are observed? These are other Raman peaks of MB molecules)? Or the Raman peaks of Si, or others.

Responses: Based on the Reviewer’s comments, we have clarified this section. No other Raman shifts were observed in the SERS spectrum; only changes in intensity were observed. Figure 4 shows the Raman spectrum of MB in ethanol. The peaks at 883 and 1067 cm-1 were attributed to the characteristic peaks of ethanol. The unmarked peak at 520 cm-1 in Figure 5 was attributed to Si, and the other peaks were attributed to the Raman peaks of MB and are listed in Table 1.

7. Please read 2 or 3 times your manuscript, because it remain some spelling errors.

Responses: Based on the Reviewer’s comment, the revised manuscript has been edited for proper English language, grammar, punctuation, spelling, and overall style by two of the highly qualified native English-speaking editors at American Journal Experts (www.aje.com).

The editorial certificate for the revised manuscript is attached as supporting information for the Editor.

Round 2

Reviewer 4 Report

After reading the revised version, I am OK with the fact that the authors have improved their paper. However, I did not change my opinion on the fact that this paper is relatively out of scope of this journal. Moreover, I asked to put at least a SEM image with a low magnification, and no image. The interest of this image is to see the assembly of nanospheres on a surface which presents several domains showing a non-uniformity of assembling on large surfaces. I also asked to put the values of the enhancement factor in the main text and not in Supplementary Informations (OK for calculations), and no discussion is presented in this paper about this value of EF. Last thing, you tell me that in figure 5(a), all the Raman peaks displayed in these SERS spectra are Raman peaks of MB molecules except the peak at 520 cm-1 (due to the presence of Si), but it is wrong: indeed, this peak at 520 cm-1 is well a peak of Si but also in the domain from 920 cm-1 to 1000 cm-1, this group of Raman peaks corresponds to Raman peaks of Si (Multi-phonons peaks of Si, see for instance this reference: Khorasaninejad et al, NANOTECHNOLOGY 2012, 23, 275706) and not Raman peaks of MB molecules. Thus, for all these reasons, I reject this paper.

Author Response

We appreciate the Reviewer's comments on this manuscript, which have helped us sbustantially revise the manuscript.

All the replies are written in boldface. According to the valuable comments of Reviewer, we  prepared a highlighted revised version documenting all changes made.

Our point-by-point response to the reviewer’s comments is included with the attached file.
